# Properties of the Spanish Version of the Place Standard Tool

**DOI:** 10.3390/ijerph19159395

**Published:** 2022-07-31

**Authors:** Ana Ocaña, Vicente Gea-Caballero, Rosana Peiró, Raúl Juárez-Vela, Elena Pérez-Sanz, Silvia Corchón, Joan Josep Paredes-Carbonell

**Affiliations:** 1Inequality in the Health Area, Foundation for the Promotion of Health and Biomedical Research in the Valencian Region (FISABIO), 46020 Valencia, Spain; a.ocanao@gmail.com; 2Local Action and Health Equity Group, Foundation for the Promotion of Health and Biomedical Research in the Valencian Region (FISABIO), 46020 Valencia, Spain; peiro_ros@gva.es (R.P.); perez_elesan@gva.es (E.P.-S.); paredes_joa@gva.es (J.J.P.-C.); 3Public Health Centre of Gandia, Gandia, 46701 Valencia, Spain; 4Faculty of Health Sciences, VIU Valencia International University, 46006 Valencia, Spain; 5Research Group SALCOM Community Health and Care, Valencia International University, 46006 Valencia, Spain; 6General Directorate of Public Health and Addictions, 46020 Valencia, Spain; 7CIBERESP ISCIII, 28029 Madrid, Spain; 8Department of Nursing, University of La Rioja, 26006 Logroño, Spain; raul.juarez@unirioja.es; 9GRUPAC, University of La Rioja, 26006 Logroño, Spain; 10Public Health Centre of Valencia, 46024 Valencia, Spain; 11Nursing Department, University of Valencia, 46010 Valencia, Spain; silvia.corchon@uv.es; 12Ribera Health Department, Alzira, 46600 Valencia, Spain

**Keywords:** social determinants of health, community participation, urban health, surveys and questionnaires, validation study

## Abstract

(1) Background: The social determinants that maintain health inequalities are organized in the physical, social, and economic contexts of neighborhoods and municipalities. Their characteristics influence the behaviors and choices of the people living in them, with an impact on their health and well-being. In recent years, several local applications and urban development tools have been designed to learn how to promote the development of health and wellness environments. Aim: The purpose was to test the properties of the Spanish adaptation of the Place Standard Tool through its implementation in a Valencian community municipality. (2) Methods: Metric properties were analyzed from a sample of 242 participants. Descriptive statistics were used to analyze the sociodemographic data and to describe item responses. Cronbach’s alpha was used to provide a measure of the internal consistency, whereas the Kaiser–Meyer Olkin test was relied upon to study the relationship between different variables. (3) Results: The questionnaire showed an internal consistency index of 0.849 and a KMO of 0.842, with a single factor variance of 81.50%. (4) Conclusions: The Spanish adaptation of the Place Standard Tool is a valid tool for assessing neighborhoods and municipalities with a focus on social determinants of health and equity.

## 1. Introduction

Scientific evidence shows that people living in urban areas with disadvantaged socioeconomic characteristics have poorer health conditions. These health inequalities are determined by social factors, defined by the WHO as “the conditions in which people are born, grow, work, live, and age”, and they are attributable to unequal access to health-related opportunities and to their distribution by social class, gender, country of origin, age, and territory [1].

The social determinants that create and maintain inequalities in neighborhoods, towns, and cities are organized in physical, social, and economic contexts. The physical environment refers to the natural environment and the built environment that encompasses the planning of infrastructure and urban development; the use of public spaces, equipment, natural spaces, housing, mobility, and transport; environmental factors such as noise or pollution; and access to healthy food. The socioeconomic environment also includes aspects such as quality of employment and working conditions, family and domestic environment, public policies (education, health, and social services), security, social cohesion, and community participation. Places are spaces of coexistence where these factors interact, determining health and well-being. Therefore, the characteristics of neighborhoods and cities influence the behaviors and choices of people living in them, having positive or negative impacts on their health and well-being [2]. Actions on specific determinants, such as urban planning, transport, or housing, are effective in increasing opportunities to access a healthy lifestyle and have positive impacts on levels of emotional well-being, physical activity, healthy eating, social cohesion, and security [3,4,5,6].

Over the past few years, various assessment tools were developed to gather the impressions of inhabitants about the place where they live. Although many of these tools are based on physical characteristics and, to a lesser extent, on social and economic peculiarities, the resident participation is rather limited [7,8].

The World Health Organization (WHO), in the context of the Healthy Cities Network, published in 2020 [9] a compendium of tools that promote the development of health and wellness environments, highlighting instruments of local application and urban development. One of these tools, translated into more than a dozen languages, is the Place Standard Tool (PST) [10].

Since 2015, the Scottish government has implemented the PST throughout its territory within the scope of a national strategy to address determinants of health inequalities, based on an intervention in settings combined with community participation [11]. This tool evaluates the physical, social, and economic aspects of a territory through a questionnaire that collects the perspectives of its inhabitants, making visible the perceptions of how the characteristics of a place affect particular people or groups. In addition, the PST allows identification of the strengths and weaknesses of a territory, characterizes them, and establishes priorities for action based on specific determinants.

For the purpose of having such a tool in Spain, the PST was translated and adapted into Spanish, and its content was validated by qualitative methods [12], although its properties have not been described so far. The failure to have a tested questionnaire to assess the living environments of people, which is important and influential from a health perspective, justifies the development of this study.

Therefore, the aim of this study was to explore the metric properties of the Spanish adaptation of the Place Standard Tool (Entornos de Vida) in its implementation in a Valencian community municipality (Spain).

## 2. Materials and Methods

### 2.1. Design and Setting

An observational cross-sectional study of the metric properties of the Entornos de Vida (EdV) tool was undertaken in Denia (Spain) between 2019 and 2020.

The study was part of the XarxaSalut strategy [13] to study the health situation analysis stage in Denia. Following the approval by the town council and authors of the original tool, the piloting of the EdV tool took place with a view to evaluate the physical, social, and economic contexts of Denia.

Denia is a coastal municipality of the Alicante province. It has a population of 42,827 inhabitants, of which 22.7% are of foreign origin, with an average gross income of EUR 23,781. It has more than 21,000 tourist accommodation places and 450 restaurants. It is thus worth emphasizing that the urban development of the city is closely associated with residential tourism [14].

### 2.2. Participant Selection and Data Collection

The criteria for inclusion in the study were as follows: (i) to be living or working in the municipality at the time of the evaluation; (ii) to be over 18 years of age; and (iii) to expressly consent to participate in the study.

The design definition of the participation was based on three assumptions: (a) to include the equity approach in health [15,16]; (b) to count on the participation of the local administration, associated and non-associated citizenship, and professional/technical resources that manage services, programs, or benefits of the assessed municipality; and (c) to encourage the participation of the community in general.

To this end, three participation approaches were organized: (a) discussion groups, (b) field interviews, and (c) a self-administered online questionnaire.

The equity approach was applied to form the discussion groups using convenience sampling, with the aim of obtaining perceptions about items according to different axes of inequality: age, gender, functional diversity, socioeconomic level, migration status, and territory. The design of the groups and their uptake was based on the demographic characteristics of the resident population and relied on the participation of technical municipal employees and social agents of the municipality.

The sample that took part in the study through field interviews was captured during a highly attended community event. To promote online participation, a QR code that led to the online questionnaire was created. The code was included in the project outreach material and promoted through municipal social media (Facebook, Twitter, and Instagram) and community communication networks (municipal public announcements, WhatsApp, and information points in local businesses).

### 2.3. Measures

#### 2.3.1. Demographic and Socioeconomic Characteristics

At the time of participation in the study, regardless of the participation mode, and in addition to the EdV tool, all participants were requested to provide sociodemographic characteristics: gender, age, level of education, employment status, degree of disability, nationality, and neighborhood of residence or work.

#### 2.3.2. Place Assessment

The EdV tool was used to analyze the physical, economic, and social contexts of Denia [12]. The instrument is based on social determinants of health at the urban level, and its design facilitates the participatory analysis of a defined territory. It allows the identification of its strengths and weaknesses, and of how the most vulnerable groups are affected by their own traits/characteristics. It promotes structured discussions on 14 cross-linked issues that impact health and well-being (Table 1), and gathers the meaning of the characteristics in the location of its residents, defining how much it affects them unevenly, and in which ways. Each item comprises a brief description, a main question, and several secondary questions to encourage further consideration (Table A1 in Appendix A). Each item receives a score from 1 to 10, with 1 being the least favorable and 10 being the most favorable. At the end of the questionnaire, the score is graphically transferred to a spider diagram, showing which topics receive the best and worst evaluations (Figure 1).

### 2.4. Data Analysis

As for the sampling, according to Iacobucci and Duhachek [17], the sample size must be related to the length of the scale. The sample was recruited in the year 2019. For the exploratory factor analysis, Rouquette and Falissard [18] recommend 5 and 10 subjects per item, with at least 200 subjects in total. As EdV included 14 items, a minimum of 140 subjects were needed. To measure the reliability of the scale, Cronbach’s alpha was used. The Kaiser–Meyer Olkin (KMO) test was used to assess the relationship between variables. The explained variance of the questionnaire was calculated. Sociodemographic variables were summarized with descriptive statistics (mean and standard deviation in quantitative variables; frequencies and percentages in categorical variables). Descriptive statistics were used to describe item responses and to summarize scale scores. For each item, quantitative variable were calculated by summing the scores obtained for each question and taking their arithmetic mean.

Data analysis was performed using SPSS and IBM SPP-AMOS V24 (IBM Corporation, New Orchard Road Armonk, New York, NY, USA).

### 2.5. Ethical Review

Before data collection began, the study protocol was reviewed and approved by a local research ethics committee (reference no. 1208176). All participants were fully informed about the aims of the study and signed the consent form prior to completing the research instruments, and the anonymity and confidentiality of the information were maintained. This study adhered to European and Spanish data protection regulations (Organic Law 3/2018 and General Data Protection Regulation [EU] 2016/679).

## 3. Results

The study involved 242 people, 91 in discussion groups, 83 field interviews, and 68 online interviews. The participants were aged between 18 and 95 years (mean: 48.32 years; standard deviation (SD): 17.09); the majority were women (62.81%), 17.51% had a nationality other than Spanish, and 77.28% had completed middle or higher education. The districts with the most participants were Saladar, El Montgo, and Puerto/Centro (16.8%, 14.9%, and 11.9%, respectively). Table 2 describes the sociodemographic characteristics of the participating population.

Regarding the scores given to the 14 EdV items (Figure 2), the aspects related to mobility (Walking or Cycling, Public Transport, Traffic and Parking) received the worst scores (5.91, 4.04, and 4.38, respectively), along with Work and Local Economy (4.41). Other components of the built environment (Housing and Community, Care and Maintenance) also obtained low scores (5.21 and 5.28, respectively). The most valued item was Identity and Belonging (7.39), followed by Facilities and Amenities (6.87), Feeling Safe (6.80), and Natural Space (6.43).

The internal consistency of the EdV questionnaire was measured with Cronbachs’s alpha reliability coefficient, obtaining a result of 0.849.

The degree of relationship between derived quantitative variables, measured with the KMO test, had a result of 0.842.

Given the high reliability obtained, the percentage of explained variance was assessed. With a percentage of explained variance of 81.50%, the tool can be considered as a single factor.

## 4. Discussion

Our study aimed to measure the structural validity and reliability of a newly validated tool using qualitative methods [19] from another original tool (Place Standard) that, despite being produced and translated into multiple languages, had not been validated in any country, language, or culture in the world, even though it was used to assess territories [10]. This is a necessary study as it reinforces the evidence and reliability of a tool with great international acceptability—the PST, validated in Spain as EdV, which has been proposed by the WHO itself as a tool of interest to promote healthy living environments that will improve both the health and well-being of communities [9].

### 4.1. Validation of the EdV Tool

With regard to the results obtained, it can be emphasized that the true strengths of the present study are that it is possible to determine that the EdV turned out to be highly reliable. In terms of its properties, we consider that a reliability rate of 0.849 is a good result, in line with most sources that consider any score between 0.7 and 0.9 as a good internal consistency for a one-dimensional scale [20]. This result, which quantifies the correlation between the scale items, underpins its structural validity. To validate the degree of relationship between the variables, the KMO test was used, the result of which (0.842) allows us to claim that the variables of EdV are highly related to one another [21]. Finally, regarding the explained variance, we consider that the percentage of 81.50% is sufficiently high to allow the acceptance of the unifactorial structure of the tool.

As the minimum sample calculated based on the background was 200 participants, and we had a final sample of 242, we gathered a sample that allows us to have confidence in the results produced. This element is essential in this type of study, as the usual conditions are the minimum number of participants, as well as the representativeness of the sample. It is important to emphasize that, the more real and diverse the participation is during data collection with a view to assessing territories, the more visible the health of specific groups, especially vulnerable ones. By doing so, concrete intervention plans can be built for each living environment, which will enable actions to be taken based on the determinants of health inequalities [10].

### 4.2. Place Assessment

In this study, the items Walking or Cycling, Public Transport, and Traffic and Parking—which include aspects such as having suitable areas for walking and cycling; affordable, regular, and well-connected public transport; and adequate traffic and parking regulation—have obtained the worst global scores. Different studies have shown that negative impacts of aspects related to mobility, such as accessibility to public transport or dependence on cars, generate significant health inequalities [22]. As community interventions in these areas are more effective than individual interventions to promote health [23], transport planning policies are required to transform urban environments into healthy, equitable, and sustainable places [24].

The item Identity and Belonging, which refers to the history, culture, and sense of community, had the best overall score. Integrated as a dimension of social neighborhood environment, it is related to the practice of physical activity, play, active transport, and mental health [25].

The combination of assessment tools related to environments with other instruments that focus greater attention on analyzing how lifestyles are affected by specific characteristics of neighborhoods or municipalities [26,27,28,29,30] makes it easier to identify which factors of the social, physical, and economic contexts require effective and concrete strategies aimed at improving the health and well-being of the affected community.

The main limitation of the present study is not comparing the 14 items of the EdV Toolamong groups of participants. Likewise, as it is a tool that is first validated quantitatively, and there are no other similar tools with which to compare the results, the data will have value in the future, but it is not possible to compare or discuss the results with previous backgrounds of the same nature. On the other hand, the fact that it is a short tool has caused it to be one-dimensional, which is nonetheless positive as it simplifies its structure and yields positive metric results. In future studies, it would be interesting to focus on the differences in place assessment according to sociodemographic characteristics.

As a strength, it should be noted that a readily available instrument adapted to the Spanish population that can be administered in a local context is useful to collect the perspectives of citizens about their neighborhoods and municipality.

## 5. Conclusions

The Spanish adaptation of the Place Standard Tool, Entornos de Vida, is a reliable and valid tool in Spanish to evaluate neighborhoods and municipalities with a focus on social determinants of health and equity. Its use in the preparation of health situation reviews will allow the participation of citizens in actions aimed at addressing the impacts that living environments have on their health.

## Figures and Tables

**Figure 1 ijerph-19-09395-f001:**
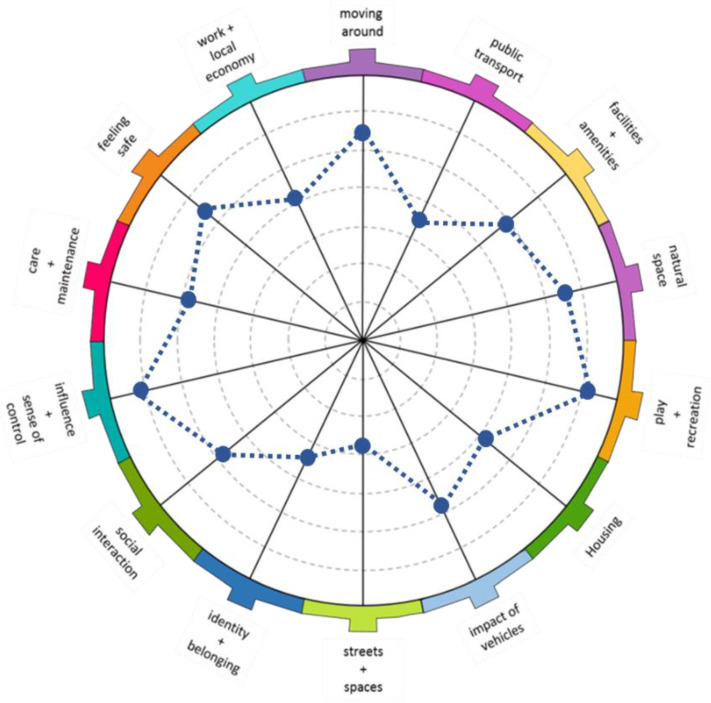
Spider diagram.

**Figure 2 ijerph-19-09395-f002:**
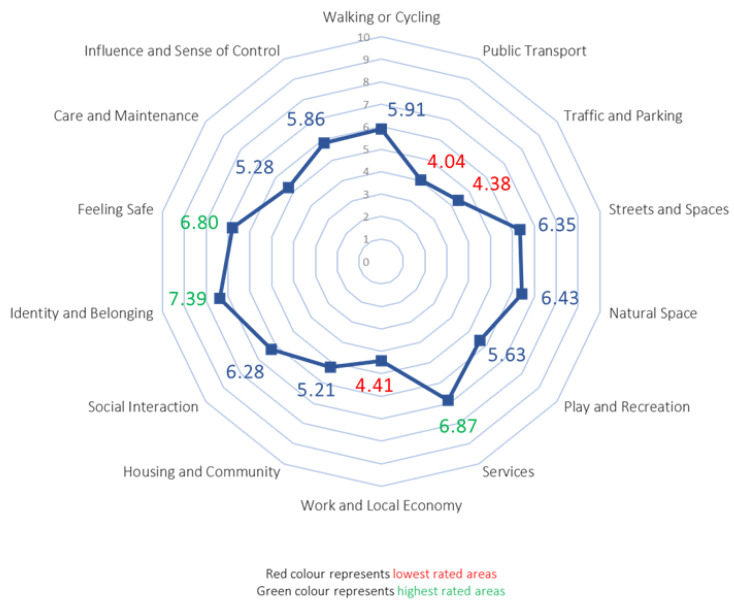
Summary of the mean scores obtained in the 14 items of the EdV tool.

**Table 1 ijerph-19-09395-t001:** The 14 topics of the physical, social, and economic environment assessed using the EdV tool.

Themes Evaluated with *Entornos de Vida*
Walking or Cycling
Public Transport
Traffic and Parking
Streets and Spaces
Natural Space
Play and Recreation
Services
Work and Local Economy
Housing and Community
Social Interaction
Identity and Belonging
Feeling Safe
Care and Maintenance
Influence and Sense of Control

**Table 2 ijerph-19-09395-t002:** Sociodemographic characteristics of the sample.

Variables		n	%
*Gender (n = 242)*			
	Female	152	62.81
	Male	89	36.78
	Other	1	0.41
*Age (n = 241)*			
	18 to 35 years	67	27.80
	36 to 50 years	61	25–31
	51 to 65 years	71	29.46
	65 or above	42	17.43
*Education Level (n = 242)*			
	Primary school or below	19	7.84
	Secondary school uncomplete	36	14.88
	Secondary school complete	86	35.54
	Tertiary education or above	101	41.74
*Employment status (n = 242)*			
	Student	30	12.40
	Employed	119	49.17
	Unemployed	25	10.33
	Unpaid care	4	1.65
	Retired	64	26.45
*Degree of Disability (n = 242)*			
	33% or below	226	93.39
	33% or above	16	6.61
*Nationality (n = 217)*			
	Spanish	179	82.49
	European Union	20	9.22
	Not European Union	18	8.29
*Neighborhood (n = 202)*			
	Les Roques	11	5.45
	Les Marines	16	7.92
	Les Rotes	9	4.46
	La Xara	12	5.94
	Baix La Mar	15	7.43
	Port/Centro	24	11.88
	La Faroleta	9	4.46
	Saladar	34	16.83
	París-Pedrera/Camp Roig	23	11.39
	Oest-Campaments	19	9.41
	El Montgó	30	14.85

## Data Availability

Data are available upon request from the author. Data are not publicly available owing to privacy and ethical concern.

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
