# Peer review of "Properties of the Spanish Version of the Place Standard Tool"

_ijerph, 2022, doi:10.3390/ijerph19159395_

Round 1
Reviewer 1 Report
Thank you for the opportunity to review this paper.
I found the paper interesting to read and believe it has important information which would be useful for policy making.
I recommend the following revisions.
1. Lines 90 to 92: The word “descriptive” is not necessary in that sentence. The authors are describing the design and setting in that section not the analysis. It seems using a verb would be more appropriate. Perhaps rephrase as:
“An observational cross-sectional study of the psychometric properties of the Entornos de Vida (EdV) tool was undertaken in Denia (Spain) between 2019 and 2020.”
2. Lines 93/94: I am not sure what this sentence means. Would it be:
“The study was part of the XarxaSalut strategy [13] in to analyse the health situation analysis
stage in Denia.”
3. Lines 99 to 100: Please delete as per below.
“The main activity sector is the one of services, tourism being the main economic activity.”
4. Line 107: The word “design” doesn’t seem appropriate in that sentence. Perhaps rephrase as:
“The design definition of the participation was based on three assumptions: …”
5. Line 112: Please replace “conversation” by “discussion”.
6. Line 114: Please rephrase as:
“The equity approach was applied to form the discussion groups by using convenience sampling, …”
7. Lines 144 to 146: In Table 1, there are 19 themes but in Figure there are 14 themes. Why? Also, In Figure 1, the definitions are in Spanish. Could the authors translate to English?
8. Lines 148/149: Please rephrase as:
“As for the sampling, according to Iacobucci and Duhachek [17], the sample size must be related to the length of the scale.”
9. Line 149: Please replace “to” by “for”.
1. Line 152: Please use the word “assess” instead of “study".
“The Kaiser Meyer Olkin (KMO) test was used to study assess the relationship between variables”
1. Line 156: Please add “and percentages” after “frequencies”
1. Lines 157 to 159: This sentence is confusing. It contains too much information. Please simplify. I assume the authors were attempting to say something along those lines?
“For each item, quantitative variables were calculated by summing the scores obtained for each question and taking their arithmetic mean”
1. Line 177: It is confusing to have “,” to indicate decimal places. Please use “62.81” instead of “62,81” throughout. Please be consistent in reporting the number of decimal places; in this case 2 decimal places throughout. Typo in “Secundary school uncomplete”. Some of the variables total don’t add up to 242 (e.g. Age, Nationality, Neighbourhood), please add missing (if any) under each variable as a separate row in Table 2.
1. Line 180: Same remark as above regarding using “,” for decimal places. Please change throughout.
1. Lines 181 to 186: Is there a cut off score in the literature to support that score of say 5 and below can be classified as worst? If so, please provide a reference.
1. Line 189: The word “studied” in “The degree of relationship between studied variables…” doesn’t seem appropriate. Perhaps “derived quantitative”.
1. Line 191: The word “studied” in “the percentage of explained variance was studied” doesn’t seem appropriate. Perhaps “assessed”.
1. Lines 192/193: This sentence is rather difficult to comprehend. Could the authors please rephrase? Perhaps start with “With a percentage explained variance of 81.50%, the tool can be considered as a single factor”. It’s also rather strange that the authors went about doing the EFA but then stating “it was not considered appropriate to perform an exploratory or confirmatory factorial analysis.” Please clarify.
1. Line 215: Please rephrase “and that finally there were 242 of them” as “and we had a final sample of 242”.
Author Response
Thank you for considering reviewing this manuscript. We consider that this review has been exhaustive and meticulous, and your contributions have improved it.
Please find attached the point-by-point response document.

Reviewer 2 Report
This is a very important study, as it uses, for the first time, the Place Standard Tool which is recommended by WHO, in a population study. This tool can reflect the needs and assets of a certain neighborhood and direct interventions.
In my opinion, the authors present here a very small part of their work and missed some of the main outcomes.
1. As they refer to the social determinants of health as indicators of inequality in health – they should have used their socioeconomic variable and compare the EdV scores of different neighborhoods, level of education, employment status, etc.
2. In table 1 they present the 14 dimensions of the tool with additional headings (movement, space, resources, civic, Stewardship) of groups of dimensions. I would expect to have explanation of these grouping and reliability tests for each one of the 5 groups. In my opinion, it is a brilliant idea to collapse dimensions for the analysis, but this should be accompanied by a reliability test.
3. The authors claim that they have a representative sample of people of Denia. In order to convince the readers, they should have compared, at least, the distribution of Denia's population by age and gender with that of the sample. The sample seems to have high proportion of women and high proportion of retired. We can not assume what its influence on the scores.
4. The title of the paper is misleading. I suggest to use the term "metric" instead of "psychometric".
Author Response

(The authors gave the same response as above.)

Round 2
Reviewer 2 Report
Unfortunately, I got only partially convincing answers to my comments.
Most of the corrections which have been made were according to reviewer 1 who commented only on editing (the corrections not always improved the text).
My first comment was about not using the demographic data to compare the Place scores. I received an answer that the demographics were not included in the questionnaire. However, in the text it is written:
"2.3.1. Demographic and Socioeconomic Characteristics
According to the axes of inequality, sociodemographic data were collected as variables: gender, age, level of education, employment status, degree of disability, nationality,
and neighborhood of residence or work."
What is the source of these data? How did they get the information? This is not mentioned at all.
The fact that we don't know how the demographic data was collected must be corrected. However, the fact that the dimensions can't be compared between groups should be mentioned in the discussion as the main limitation of the study.
My second comment referred to the categorization of the 14 dimensions of the tool into 5 categories without any justification for that (reliability test). The authors claim that they checked only the overall reliability of the 14 and even didn't try to refer to my comment. In fact, the 14 dimensions tool was built in purpose as separate dimensions, as each one measure a specific aspect. The beauty of the tool is the figure which can give, in a glance, information about a place. This way of presentation can be used for follow-up after implementation of interventions. I think that there are two possible ways to improve that part of the paper: a. To add reliability tests to the 5 categories, or b. to present the 14 dimensions as they are without mention the categories. A figure would have been much more convincing than table 3.
Thanks for deleting a sentence according to my 3rd comment.
Finally, as for my 4th comment – the title was changed, but in the text there is still "psychometric properties" – these should be corrected.
Author Response
After carrying out a careful review of our manuscript, we proceed to send it for a new evaluation. In the revised version of the manuscript we have highlighted in red the modifications made to the original text.
We want to express our sincere thanks to the reviewers and the editorial team for their time and great review work. Your comments have given us the opportunity to significantly improve the original manuscript and to reflect on future researches related to the continued study of the tool.
In the attached document, we have included the answers to all the comments made by the reviewers, point by point, as well as the changes they have produced in the manuscript.
